# Hierarchical Fusion of Infrared and Visible Images Based on Channel Attention Mechanism and Generative Adversarial Networks

**DOI:** 10.3390/s24216916

**Published:** 2024-10-28

**Authors:** Jie Wu, Shuai Yang, Xiaoming Wang, Yu Pei, Shuai Wang, Congcong Song

**Affiliations:** 1Changchun Institute of Optics, Fine Mechanics and Physics, Chinese Academy of Sciences, Changchun 130033, China; 15754314374@163.com (J.W.); 13604311978@163.com (X.W.); peiyu@ciomp.ac.cn (Y.P.); 18704456420@163.com (C.S.); 2Suzhou Institute of Biomedical Engineering and Technology, Chinese Academy of Sciences, Suzhou 215163, China; wangshuai@sibet.ac.cn

**Keywords:** image fusion, guided filter, generative adversarial network, histogram mapping, channel attention mechanism

## Abstract

In order to solve the problem that existing visible and infrared image fusion methods rely only on the original local or global information representation, which has the problem of edge blurring and non-protrusion of salient targets, this paper proposes a layered fusion method based on channel attention mechanism and improved Generative Adversarial Network (HFCA_GAN). Firstly, the infrared image and visible image are decomposed into a base layer and fine layer, respectively, by a guiding filter. Secondly, the visible light base layer is fused with the infrared image base layer by histogram mapping enhancement to improve the contour effect. Thirdly, the improved GAN algorithm is used to fuse the infrared and visible image refinement layer, and the depth transferable module and guided fusion network are added to enrich the detailed information of the fused image. Finally, the multilayer convolutional fusion network with channel attention mechanism is used to correlate the local information of the layered fusion image, and the final fusion image containing contour gradient information and useful details is obtained. TNO and RoadSence datasets are selected for training and testing. The results show that the proposed algorithm retains the global structure features of multilayer images and has obvious advantages in fusion performance, model generalization and computational efficiency.

## 1. Introduction

Infrared and visible image fusion is an important technique in multimodal image fusion, aiming at fully integrating the most meaningful and valuable information extracted from infrared and visible sensors, respectively, to generate a fused image with richer information and higher quality [1]. Visible images contain more detailed information, such as chromaticity and saturation, but they are easily affected by environmental factors, such as haze and illumination [2,3,4]. An infrared image is a thermal image of an object that produces clear foreground outline information, even under the influence of low light and other environmental factors, but contains fewer pixels, resulting in insufficient detail information [5]. The fusion of infrared images and visible light images can compensate for their respective disadvantages; the advantages of the image will be concentrated to produce informative images. This image fusion technology has a wide range of applications in the field of target detection, object recognition and military detection [6].

The advantage of fusing infrared and visible images is that it contains both the clear texture details of visible images and the anti-interference properties of infrared images [7,8,9]. There are roughly three types of image fusion methods: one is the traditional transformation domain method, which mainly includes the Laplace Pyramid fusion method [10,11,12], Guided Filter [13,14] and Non-Subsampled Contourlet Transform (NSCT) [15]. This type of algorithm analyzes the image through filters for the features of the image to achieve fusion, with the disadvantage of over-reliance on complex fusion rules. The second is a fusion method based on the spatial domain [16,17]; this type of method downscales the dataset source image information, with the disadvantage being that the amount of computation is large. The third is a fusion method based on deep learning, mainly using convolutional neural network deep extraction of image features in preparation for the subsequent fusion stage. The disadvantage is that when acquiring image features, it is affected by network depth and other factors, and the fused image often suffers from information loss [18,19,20]. Starting in 2018, the Generative Adversarial Network (GAN) [21] began to take an advantage in image fusion. This method trains the network using infrared and visible images as training sets. The network includes a generator and a discriminator to extract features and guide image fusion, thereby obtaining a trained network model to fuse infrared and visible images. The GAN algorithm avoids complex fusion rules, but the random network initialization parameters make the results of the adversarial generation stage uncertain [22,23]. In the existing fusion algorithms based on GAN, it is difficult to get rid of the dependence on content loss. Some algorithms introduce a large amount of noise during the fusion process, making the fused image extremely inconsistent with human visual perception and severely distorted [24,25].

In view of the above problems of infrared and visible image fusion, this paper proposes a multilayer fusion method based on improved GAN and channel attention mechanisms. Firstly, the infrared and visible images are decomposed into a base layer and a refinement layer, respectively, by a guided filter, and the base layer of the visible image is enhanced by histogram mapping for contour processing. Then, the base layer of the two is fused by the Laplace fusion method. Then, a fusion network is designed by adding a guided fusion module and a depth transferable module based on the GAN to fuse the refinement layer of the two and increase the extraction of detailed features from fused images. Finally, a multilayer convolutional fusion network is designed based on the channel attention mechanism for improving the fused image saliency information, and the layered processed images obtained in the first two steps are fused into the final image. The resulting image retains the local texture of the visible image and the global structural features of the infrared image, so as to make the image have a natural look and feel.

## 2. Basic Principle

GAN is an unsupervised learning method that mainly consists of two parts: *G* (Generator) and *D* (Discriminator). The basic principle of GAN is through the adversarial training of the generator and discriminator so that the generator is able to generate data close to the real data distribution [26]. The input of the generator *G* is a random noise vector z, and the output is the data $G\left(z\right)$, which tries to mimic the real data distribution; the discriminator *D* is for judging the relevance of the real dataset and the data x generated by the generator [27]. The loss function is a key component in evaluating the performance of the generator and the discriminator. The loss function in GAN is designed to maintain a balance between the generator and the discriminator so that they can effectively work against each other and progress together.

The loss function is defined as follows:(1)$$V(D,G)=E_{x\sim P_{data(x)}}[\ln D(x)]+E_{z\sim p_{z(z)}}[\ln(1-D(G(z)))]$$
where $E_{x\sim P_{data(x)}}$ represents the actual input value of the sample used in training, x represents the training set of real images used for training, $E_{z\sim p_{z(z)}}$ refers to the samples extracted from known noise, $P_{data}$ represents the image distribution and $P_{z}$ represents the noise distribution. The generator and discriminator continuously learn and optimize model parameters through gradient descent and ultimately fuse the model.

## 3. Improved Image Fusion Algorithm Design

This paper relates a novel method for the fusion of visible and infrared images, and the proposed fusion framework is shown in Figure 1.

The guided filtering algorithm is a local linear model that employs a guided image I to perform a filtering operation on the input image P to enhance the edge features while smoothing the input image with the following formula:(2)$$W=G_{r,\varepsilon }(P,I)$$
where W is the output image, I denotes the guided image, r denotes the filter window size, which determines the ability of filtering—the smaller the value, the less smooth the filtered output will be—and ε is a regularization coefficient—the smaller the value, the higher the edge retention of the image. Compared to traditional methods, the multiscale decomposition technique using the guided filter can separate the overlapping features in the image null domain and decompose the source image into two dimensions. Assuming $I_{k}\left(k=1,2\right)$ is the input images, then $I_{1}$ and $I_{2}$ are the infrared image and visible image, respectively. For each input image, the base layer can be obtained by solving Equation (3), as follows:(3)$$I_{k}^{b}=\arg\min||I_{k}-I_{k}^{b}|{|}_{F}^{2}+\lambda (g_{x}*I_{k}^{b}|{|}_{F}^{2}+||g_{y}*I_{k}^{b}|{|}_{F}^{2})$$
where $g_{x}=[1-1]$ denotes the horizontal gradient operator, $g_{x}={\left[1-1\right]}^{T}$ denotes the vertical gradient operator, and λ=5. The refinement layer $I_{k}^{d}$ is the part of the base layer subtracted from the source image, which is defined as follows:(4)$$I_{k}^{d}=I_{k}-I_{k}^{b}$$

Since the base layer contains large-scale contour information, this paper designs the use of histogram mapping and the Laplace algorithm to enhance the low-frequency information of the visible base layer and fuse it with the infrared base layer. The refinement layer contains small-scale texture information and edge information, and this paper designs a fusion network based on GAN by adding a bootstrap fusion module and a depth relocatable module to fuse the infrared and visible refinement layers. A multilayer convolutional fusion network is designed based on the channel attention mechanism to fuse the layered processed images obtained in the first two steps into the final image.

### 3.1. Laplace Fusion of Base Layer Histogram Maps

#### 3.1.1. Visible Base Layer Histogram Mapping

As shown in Figure 2, the visible image base layer $I_{1}^{b}$ uses sub-histogram mapping to enhance the contrast of low-light images, resulting in a visible enhancement of the base layer $I_{1}^{bh}$. The fusion rule of the basic layer adopts the Laplace transform to fuse $I_{1}^{bh}$ and $I_{2}^{b}$. The method of histogram mapping is to use the local minimum to divide the image *S* into m×n subintervals and to enhance the basic image T through the subinterval histogram.

We use local entropy to control the mapping range of sub-histograms to avoid excessive histogram peaks produced by traditional methods. Where the local entropy is used to measure the texture richness of the image block, if the local entropy is smaller, it represents less texture, which will be remapped to a larger range. The mapping range controlled by local entropy is defined as follows:(5)$$R_{j}=\frac{{\left(T_{j}\right)}^{\gamma_{j}}}{\sum_{j}{\left(T_{j}\right)}^{\gamma_{j}}}$$
where $\gamma_{j}\in [0,0.8]$ is used for normalization, and $T_{j}=\sum_{k=m_{j}}^{m_{j+1}}p\left(I_{1}^{bk}\right)$ is the cumulative distribution function of the sub-histogram interval $\left[m_{j},m_{j+1}\right]$. When there are fewer texture intervals in the histogram, $R_{j}$ will be allocated to more mapping intervals. Therefore, the final visible enhancement of the base layer $I_{1}^{bh}$ is calculated as:(6)$$I_{1}^{bh}=\sum_{t=1}^{j-1}R_{t}+R_{j}\cdot \frac{\sum_{k=m_{j}}^{I_{1}^{bk}}p\left(I_{1}^{bk}\right)}{\sum_{k=m_{j}}^{m_{j+1}}p\left(I_{1}^{bk}\right)}$$

#### 3.1.2. Base Layer Laplace Fusion

The fusion process of the image base layer is shown in Figure 3. Where $L{\left\{I_{1}^{bh}\right\}}^{l}$ and $L{\left\{I_{2}^{b}\right\}}^{l}$ represent the L-layer Laplace pyramid decomposition of visible enhancement of the base layer $I_{1}^{bh}$ and the infrared base layer image $I_{2}^{b}$, respectively, l∈(1,2,3,4).

Firstly, the fusion formula for the top-level image $L{\left\{I_{1}^{bh}\right\}}^{4}$ and $L{\left\{I_{2}^{b}\right\}}^{4}$ is as follows:(7)$$G=\frac{1}{\left(m-1)(n-1\right)}\sum_{i=1}^{m-1}\sum_{j=1}^{n-1}\sqrt{(\Delta I_{x}^{2}+\Delta I_{y}^{2})/2}$$
where m×n represents the region in the image where gradients are calculated, and $\Delta I_{x}$ and $\Delta I_{y}$ represent the first-order difference of pixel f(x,y) on the *x*-axis and *y*-axis, respectively. $G_{1}(i,j)$ and $G_{2}(i,j)$ represent the average gradient of each pixel in the top layer, which can reflect the clarity of the image. The top-level image fusion is defined as follows:(8)$$L{\left\{F_{b}\right\}}^{4}(i,j)=\left\{\begin{array}{l} L{\left\{I_{1}^{bh}\right\}}^{4}(i,j){\;G}_{1}(i,j)\geq G_{2}(i,j)\\ L{\left\{I_{2}^{b}\right\}}^{4}(i,j){\;G}_{1}(i,j)<G_{2}(i,j) \end{array}\right.$$

Then, the remaining four layers of Laplace fusion require calculating the regional energy $I_{k}^{b}RE(i,j)$ of each layer in the infrared and visible images,
(9)$$I_{k}^{b}RE(i,j)=\sum_{-p}^{p}\sum_{-q}^{q}w(p,q)|L\{I_{l}^{b}{\}}^{l}(i+p,j+q)|$$
where w is a 3×3 matrix, and p and q are both set to 1. When 0<l<5, the fusion result of the l layer is as follows:(10)$$L{\left\{F_{b}\right\}}^{l}(i,j)=\left\{\begin{array}{l} L{\left\{I_{1}^{bh}\right\}}^{l}(i,j)\;I_{1}^{b}RE(i,j)\geq I_{2}^{b}RE(i,j)\;\\ L{\left\{I_{2}^{b}\right\}}^{l}(i,j)I_{1}^{b}RE(i,j)<I_{2}^{b}RE(i,j) \end{array}\right.$$

Finally, $L{\left\{I_{1}^{bh}\right\}}^{l}$ and $L{\left\{I_{2}^{b}\right\}}^{l}$ reconstruct to obtain the fused image of the base layer.

### 3.2. Thinning Layer Fusion Algorithm

#### 3.2.1. Network Design

In this paper, the refinement layer fusion uses the infrared image refinement layer $I_{1}^{d}$ and the visible image refinement layer $I_{2}^{d}$ as training sets to train the fusion model of the improved GAN. The image fusion process is optimized and guided through adversarial learning. The improved GAN network consists of a generator, an encoder and two discriminators, as shown in Figure 4. There are two differences from the traditional generative adversarial network GAN: Firstly, an additional Guided Fusion Module (GFM), consisting of a discriminator and convolutional module, is added to provide a prior optimization model to guide the generation of fused images (See red dashed box in Figure 4). The second is to add deep feature transferable modules (DFTMs) to the generator network to deepen the extraction of deep features from the source image and learn the content of large targets in more detail to achieve the best fusion effect (See blue dashed box in Figure 4).

The purpose of the encoder is to reduce the dimensionality of the source image and provide a prior initialization constraint model. The infrared image refinement layer $I_{1}^{d}$ is fed into the encoder as input x, and the newly added encoder *E* maps the input x to a low-dimensional feature vector z through convolution operation, concatenates it with a label vector l and constrains the network parameters through low rank prior decomposition.
(11)$$E(x)=z\in R^{n}$$
where n is the dimension of the source image features, and the output preserves the texture features of input x.

The purpose of the generator is to take the extracted input image features as input and output the fused image. As shown in the red dotted box in Figure 4, the depth feature extraction can be realized through dense convolution by adding the depth feature migration module based on the network. The deep feature transfer module uses dense connections to extract the features of real semantic information. Assuming that the network has L layers, $x_{0}$ is the input of the network, $x_{l}$ is the output of the l layer in the network, $x_{l-1}$ is the output of the l−1 layer and H() is the nonlinear transformation acting on the l layer; the structure is shown in Figure 4, and the relationship is shown as follows:(12)$$x_{l}=H_{l}(\left[x_{0},x_{1},\ldots ,x_{l-1}\right])$$

By using skip connections to integrate the features of each stage and adding the output features of the previous layer to each input, the dimension can be expanded. In the generator, two deconvolution methods are used to map low-dimensional images to high-dimensional images. The label l is connected to z, and a new vector [z,l] is fed back to the generator. The output of the refinement layer fusion image $F_{d}$ is defined as follows:(13)$$F_{d}=G(z,\;l)=G(E(x),\;l)$$

The purpose of the discriminator is to guide the output fused image to be closer to the real image. In the fusion process of transferring information to the generator, the visible image refinement layer $I_{2}^{d}$ with richer texture features is used as the target image, a discriminator $D_{z}$ is constructed between the infrared image refinement layer $I_{1}^{d}$ and a target discriminator is constructed between the fusion image refinement layer $F_{d}$. The discriminator $D_{z}$ is used to distinguish the low-dimensional infrared detail image z generated by the encoder while forcing the distribution of the generated z to gradually approach the prior. The target discriminator is used to perform adversarial learning between the refined layer fused image $F_{d}$ and the visible detail image $I_{2}^{d}$, making the refined layer fused image $F_{d}$ more realistic. The network structure is shown in Figure 4, and each convolutional layer in the discriminator uses LRelu as the activation function, making the training process non-linear.

#### 3.2.2. Loss Function Design

The training loss functions for *E* (Encoder) and *G* (Generator) in this paper are as follows:(14)$$\underset{E,G}{min\;}L(x,\;G(E(x),\;l))$$
where L() represents the $L_{2}$ norm, x represents the infrared image refinement layer $I_{1}^{d}$, G() and E() represent the generator output and encoder output, respectively, and i represents the label vector. *E* and *G* are both updated based on the loss function $I_{2}$ between the input and output surfaces.

The loss function $L_{D_{z}}$ of discriminator $D_{z}$ is as follows:(15)$$L_{D_{z}}=\frac{1}{N}\sum_{n-1}^{N}(D_{z}(I_{2}^{d})-c{)}^{2}+\frac{1}{N}\sum_{n-1}^{N}(D_{z}(F_{d}^{n})-d{)}^{2}$$

The loss function $L_{D}$ of target discriminator D is as follows:(16)$$L_{D}=\frac{1}{N}\sum_{n-1}^{N}(D(I_{2}^{d})-c{)}^{2}+\frac{1}{N}\sum_{n-1}^{N}(D(F_{d}^{n})-d{)}^{2}$$
where b and c represent the real labels of the refinement layers $I_{1}^{d}$ and $I_{2}^{d}$, while d represents the real labels of the fusion refinement layer. $D(I_{1}^{d})$, $D(I_{2}^{d})$ and $D(F_{d}^{n})$ represent the classification results for $I_{1}^{d}$, $I_{2}^{d}$ and $F_{d}$, respectively.

### 3.3. Fusion Image Reconstruction

#### 3.3.1. Attention Mechanism Fusion Network Design

Through the above two sections, the fusion base layer images $F_{b}$ and $F_{d}$ fusion refinement layer images can be obtained. $F_{b}$ currently lacks high-contrast information, while $F_{d}$ lacks texture details in some visible light images. To address the appeal problem, an efficient channel attention mechanism is introduced to encode image $F_{b}$ and $F_{d}$, which guides the model to distinguish between salient targets and texture details. A fusion network (Dense Channel Attention Mechanism Network, dCAMN) is designed by combining a multilayer convolutional block, a downsampled convolutional block, a dense connection module and a channel attention mechanism, through which feature fusion is carried out, effectively fusing the depth features extracted by the improved GAN network and the visible texture information after histogram mapping and using the decoder to reconstruct the output fusion image $F_{last}$ so as to extract and fuse the desirable targets and texture details in each modality. The fusion method is shown in Figure 5.

As shown in Figure 5, the overall network is mainly composed of max pooling layers, 3×3 convolutional layers, and convolutional layers with activation functions that cross each other. Firstly, the max pooling layer can also better preserve the detailed texture information. After passing through the max pooling layer, the feature information is processed twice through a 3×3 convolution and a convolution layer with an activation function to enhance the detailed features and preserve the texture information. Secondly, the main body adopts dense connections using two 3×3 convolutional layers and a 3×3 convolutional layer with an LReLU activation function to concatenate blocks for feature extraction. Introducing the dense connection module Dense into the main body can fully utilize the features extracted by each convolutional layer, and using 1×1 regular convolutional layers eliminates channel dimension differences. Finally, the channel attention mechanism included in this network structure can perform feature learning in the channel dimension to form the relevant weights of the important features of each channel, which can be applied simultaneously with the features extracted from the deep convolution to better obtain the detail information of the target area that needs to be focused on, so as to achieve the demand of retaining more detail information.

#### 3.3.2. Encoder and Decoder Loss Function

The loss function $L_{s}$ is used to train the fusion network, defined as follows:(17)$$L_{s}=L_{pixel}+50L_{ssim}$$
where $L_{pixel}$ and $L_{ssim}$ represent the pixel loss and structural similarity loss between the input image and the output image. The pixel loss function $L_{pixel}$ is defined as follows:(18)$$L_{pixel=||Output-Input|{|}_{F}^{2}}$$
where $||\cdot |{|}_{F}^{2}$ is the Frobenius norm. The similarity between the output image and the input image at the pixel level is constrained by $L_{pixel}$. The loss function $L_{ssim}$ is defined as follows:(19)$$L_{ssim}=1-SSIM(Output,Input)$$
where SSIM(•) represents a measure of structural similarity, which quantifies the structural similarity between two images.

## 4. Experiment and Result Analysis

The dataset of the paper adopts the public datasets TNO [28] and RoadSence [29], which are widely used in image fusion tasks. The training set is 63,200 images, mainly including real scenes such as boats, streets, houses, woods, etc., and the test set is 13,500. This includes, in addition to the original dataset images, training data extended by simultaneously scaling and brightness-adjusting the infrared and visible images to address overfitting in image fusion and to maintain their corresponding alignment. In this paper, all scene types are used as the test sets in the quantitative evaluation, and two scenes, Streamboat and Street, are mainly used as the test sets in the qualitative evaluation.

### 4.1. Experimental Setup

#### 4.1.1. Experimental Environment

The image fusion network proposed in this paper is implemented in Python using the PyTorch 1.10.0 framework on an Ubuntu 22.04 operating system with a 3.50 GHz Intel Core i9-11900K CPU (Intel Corporation, located in Santa Clara, CA, USA) and an NVIDIA 3080 GPU (NVIDIA Corporation, located in Santa Clara, CA, USA) accelerated by CUDA 11.8. The batch training size is set to 16, and the total number of training epochs is set to 200. The model uses the Adam optimizer for optimization, with an initial learning rate set at 0.0001. Training is stopped when the model’s performance on the validation set begins to degrade, indicating possible overfitting.

#### 4.1.2. Evaluation Metrics

The experiment uses six metrics: entropy (EN), standard deviation (SD), normalized mutual information (NMI), structural similarity (SSIM), gradient-based fusion performance (Nabf), and peak signal-to-noise ratio (PSNR) to evaluate different fusion methods.

Among these, EN based on statistical features is an index to measure the richness of image information, and the value of EN is directly proportional to the quality of the fused image; SD is a scale standard to measure the uniformity of the sample distribution, and a larger value of SD indicates that the image has a higher contrast; NMI is an index to measure the similarity of the two images, and a larger value of NMI indicates that the fused image contains a greater amount of information from the source image; Nabf represents the noise rate in the fusion process, and the trend value of Nabf is inversely proportional to the fusion effect; SSIM represents the structural similarity between the fused image and the original image, and the trend value of SSIM is directly proportional to the fusion effect.

### 4.2. Ablation Experiment

In order to verify the effectiveness of the added GFM, DFTM, and dCAMN modules in the fusion network model, we eliminated the relevant components according to the following scheme, did ablation experiments for the relevant modules, and conducted experiments on the TNO dataset, respectively. The experiments were conducted using the control variable method, removing one of the modules of the improved part of the network as a training network at a time and selecting 30 pairs of images from each of the TNO datasets for testing, where No GFM stands for the training network after removing the bootstrap fusion module from the improved GAN, No DFTM stands for the training network after removing the depth migration module from the improved GAN, and No dCAMN represents the training network after removing the channel attention module. Some of the test results are shown in Figure 6. The orange boxed area on the left side of the farm map is the trunk of a tree at the edge of a house, and the orange box on the right side is a shrub; the red boxed area in the tank map represents the reflective area of the tank; the blue area in the house and tree maps is a tree branch with a sky background.

To qualitatively analyze some of the test scenes in Figure 6, when the GFM module is removed, it can be seen from the third line in the figure that the bootstrap image lacks a natural look and feel in some of the features, such as the reduced contrast of the branches of the tree partly behind the house in the third image, resulting in obvious darkness. When the DFTM module is removed, it can be seen from the fourth line in the figure that the part of the sky on the right side of the house in the first image is obviously missing, which is due to the fact that this module can fuse deeper features and effectively improve the information content of the image. When the dCAMN module is removed, it can be seen from the fifth line in the figure that there is no obvious distinction between the tail of the tank and the woods behind it in the second image, which is due to the fact that the detail layer and the edge base layer are not sufficiently fused. From the experiments, we can see that our various modules play an important role in the algorithm.

In order to do the quantitative analysis of the ablation experiments, 30 pairs of images were selected from the TNO dataset and RoadSence dataset, respectively, to do the testing, and the metrics after testing the respective images were averaged for the analysis of the metrics after removing the modules, as shown in Table 1 and Table 2.

From the data in Table 1 and Table 2, it can be seen that the use of bold markers is the best indicator value for the experiment. The GFM module improves the EN and SD indicators more obviously, which can integrate the local and global relationships and better obtain the representation of the image. The DFTM module has added the Nabf metric, enhancing the depth of feature fusion and obtaining more detailed features; the dCAMN module reduces the PSNR index, sets more attention to effective targets in the image and suppresses background noise. By comparing the results of the ablation experiment, it can be seen that each improved and added module has enhanced its original functions, verifying the effectiveness and superiority of the designed module.

### 4.3. dCAMN Module Fusion Effect Analysis

In the feature fusion stage, the dCAMN fusion network is used to increase the learning weight of effective target information, weaken background information and highlight the role of the fusion target. Figure 7 shows the comparison effect before and after adding the channel attention mechanism network, where blue represents the importance of shallow features and red represents the importance of deep features, the darker the color means the better the learning effect. stage0_Conv_features represents the first convolutional feature heatmap of $F_{d}$ and $F_{b}$ fusion, while stage1_Conv_features and stage2_Conv_features represent the output of the middle two layers of the ordinary fusion process, respectively. stage1_dCAMN_features, stage2_dCAMN_features and stage3_dCAMN_features represent the fused feature heatmap outputs from the three-layer fusion network through the channel attention mechanism. Among them, red represents the degree of importance. From the figure, it can be seen that after adding the channel attention mechanism, the network has a stronger ability to effectively fuse key features of the target location in the refinement layer and suppress areas with high confidence. The fusion in the background area produces less noise.

### 4.4. Comparative Experiment and Qualitative Analysis

Seven fusion methods of cross-bilateral filtering (CBF) [30], gradient transfer fusion (GTF) [31], DIVFusion [32], CDDFuse [33], RFN-Nest [34], GANMcC [35] and DDcGAN [36] were selected to compare with the proposed method. In order to obtain a more comprehensive evaluation, qualitative and quantitative evaluation methods were used for analysis. Among them, qualitative analysis is based on the human visual system and is the final way for the human visual perception system to express image information, but it is affected by many uncontrollable factors, such as color saturation, screen resolution, display form, environmental light, etc. This evaluation method can be one of the standards to measure the quality of fused images. It needs to be combined with objective indicators to obtain a fairer judgment. Figure 8 and Figure 9 are the comparison graphs of the results of our HFCA_GAN fusion algorithm and other mainstream fusion algorithms in the test set images on Streamboat and Street, respectively.

In Figure 8, the orange box represents the area where the landscape meets the water, and the red box represents the area of the hull detail. In Figure 9, the red box represents the area for pedestrians and the orange box represents the area for cars. From the results, it can be seen that the fused images obtained by the traditional algorithms CBF and GTF are less effective; for example, most of the visible detail information of the ship hull in the red circle in Figure 8 is not retained during the fusion process, resulting in a narrow dynamic range of the ship hull in the gray level. The fused image of this paper’s method has a clear outline of the small target, such as the standing pedestrian in the red box in Figure 9, and subjectively, the pixels near the small target of the CBF and GTF fusion results have converged to a large light spot. Although the deep learning algorithms DIVFusion, CDDFuse, GANMcC and DDcGAN solve the illumination imbalance problem and the problem of insufficient fused image features by adaptively maintaining the intensity distribution of salient targets and preserving the texture information in the background, they do not fuse the infrared targets and the visible details well, have little detail information near the hot targets, are prone to blocking, such as the orange box mountain background in Figure 8 and the vehicle headlights in Figure 9, and the CDDFuse and DDcGAN algorithms lose some of the useful detail information while counteracting the generated image. DIVFusion provides fusion enhancement methods in low light, but the background noise of the fused image is also retained. CDDFuse solves the low-light cross-modal feature extraction problem, but the overall contrast of the fused image is significantly reduced. The fused image of this paper’s method looks more natural and retains the contrast better with higher recognition.

### 4.5. Comparative Experiment and Quantitative Analysis

Quantitative evaluation methods can assess image quality through metrics independent of the observer or environment. In this paper, we use a combination of multiple evaluation metrics to compare the proposed HFCA_GAN algorithm with seven mainstream and the best fusion algorithms. The experiments use the same hardware and software configurations to ensure the standardization of variables, and six metrics are used to evaluate the different fusion methods. The metrics of the seven mainstream algorithms on the test set of image pairs from Streamboat and Street are shown in Figure 10 and Figure 11.

From Figure 10, it can be seen that our fusion algorithm has significantly higher EN values on the image pair from Streamboat than the other algorithms, which further verifies that our fused images have higher richness, and, therefore, more details of the ship’s hulls are preserved in the qualitative analysis. On the contrary, Nabf metrics are the lowest, indicating that the fused image effectively rejects noise during the fusion process. As can be seen in Figure 11, the metrics displayed in the image-to-Street show that the overall effect is similar to that on the image-to-Streamboat, while it is slightly weaker than the CDDFuse in terms of the SD value, which is due to the fact that important features (e.g., pedestrians and headlights) have been emphatically selected in the Street scene to reconstruct the fused image, and the fused image highlights the contours of the target pedestrians, suppresses the halo of lights and more accurately responds to the useful information in the image.

In order to verify the generalization of the algorithm, seven mainstream algorithms were used to perform fusion experiments on all image pairs on the TNO and RoadSence datasets, and the average values of each index were calculated and taken. The experimental results are shown in Table 3 and Table 4.

As detailed in Table 3 and Table 4, we have bolded the best indicators. It can be seen that our method has achieved good fusion performance in objectively evaluating the values of EN, SD, NMI, SSIM, Nabf, PSNR and other metrics. As seen in Table 3 and Table 4, through the fusion test of the two datasets, the method in this paper has shown a slight improvement in the metrics of EN and NMI compared to the deep learning algorithms GANMcC and DDcGAN and has a more obvious advantage over traditional algorithms such as CBF and GTF. This indicates that our algorithm has higher contrast and similarity. Our algorithm is not optimal in terms of SD metrics; this is because our method achieved a multi-scale representation of the source images and selected significant features to reconstruct the fused images. Meanwhile, the algorithm improves significantly in SSIM and Nabf metrics compared to DIVFusion and CDDFuse, which are prominent fusion algorithms in the last two years. The increase in SSIM indicates that the convolutional extraction module, along with the addition of the depth migration module, plays an important role in feature extraction and model optimization, while the introduction of the channel attention mechanism module provides more learning of the information in the important regions, and the decrease in the Nabf value and the increase in PSNR both represent the reduction of noise interference in the fusion process.

### 4.6. Network Efficiency Verification Analysis

In order to verify the computational efficiency of the fusion model, the main appeal method and the proposed algorithm HFCA_GAN were compared in the same environment. The loss function trend of the HFCA_GAN algorithm during training on two data sets, “RoadSence” and “TNO”, is shown in Figure 12.

It can be seen from Figure 11 that during the training of the two data sets, the HFCA_GAN algorithm can reach stable convergence at 100 epoch. In this paper, we selected seven current classical and cutting-edge algorithms to compare the fusion efficiency of HFCA _GAN with other fusion algorithms for inferential fusion under optimal weights. In the TNO and Roadscene datasets, we selected 20 groups of visible and infrared images under different scenes from each, performed a fusion test in the same hardware and software experimental environments, recorded the time required for the fusion process in each group, and took the average of the 20 groups of time as the fusion average time of the algorithms under the dataset. The results are shown in Table 4, and we represent the optimal value in bold. It can be seen that the HFCA _GAN algorithm achieves excellent computational efficiency, with an average fusion time of 0.203 s in the TNO dataset and 0.043 s in the Roadscene dataset. Overall, the HFCA _GAN algorithm achieves significant results in terms of fusion efficiency in both datasets, and the comprehensive performance is better than other models. See Table 5.

## 5. Conclusions

In this paper, a hierarchical image fusion method based on channel attention mechanism and improved GAN is proposed, which solves the problems of blurred edges and low sharpness of traditional visible and infrared image fusion methods. First, the image is decomposed into a base layer and a refinement layer through the guided filter, and the enhancement and fusion processes are performed on them, respectively, so that the final fused image better contains the details and contour information of the two modalities; second, the base layer of the visible image is enhanced through histogram mapping, which can improve the contour effect of the fused image; additionally, the new depth relocatable module extracts more feature information of the source image, which makes the fused image richer in detail information in the process of adversarial learning and achieves better results in SD and Nabf metrics; finally, the use of the channel attention mechanism can increase the detail texture information of the fused image of infrared and visible images, which effectively reduces artifacts and noises and retains high contrast. Through a large number of qualitative and quantitative experiments, it is shown that the method in this paper preserves the global structural features and local texture of the fused image, giving the image a natural look and feel. Future work can consider improving the attentional fusion mechanism of the base and refinement layers, and in the process of deepening the network, the edge layer will be screened for information to reduce the fusion interference of invalid targets.

## Figures and Tables

**Figure 1 sensors-24-06916-f001:**
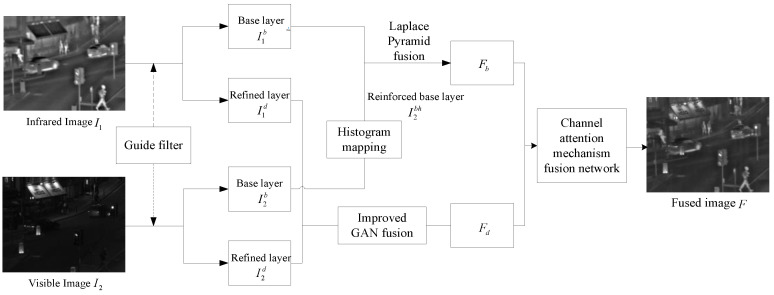
Framework of the proposed fusion method for infrared and visible image fusion.

**Figure 2 sensors-24-06916-f002:**
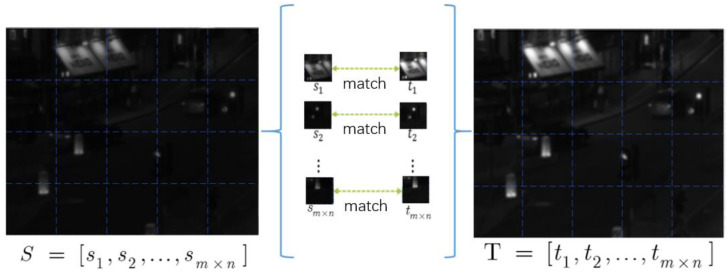
Enhanced mapping of base layer histograms.

**Figure 3 sensors-24-06916-f003:**
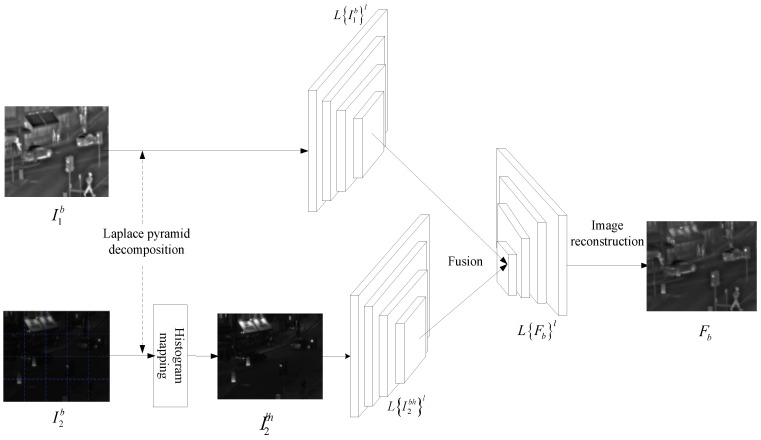
Framework for fusion of basic layer of infrared and visible images.

**Figure 4 sensors-24-06916-f004:**
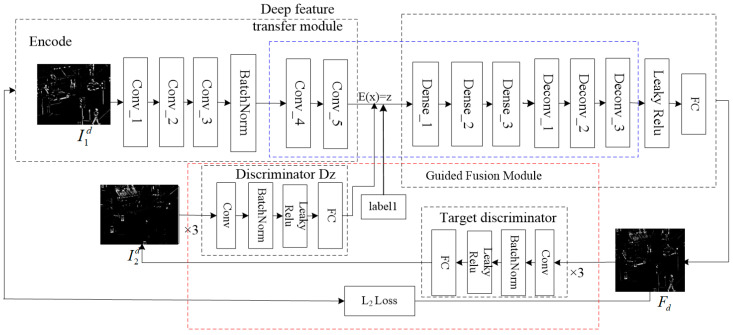
Framework for fusion of refinement layer of infrared and visible images.

**Figure 5 sensors-24-06916-f005:**
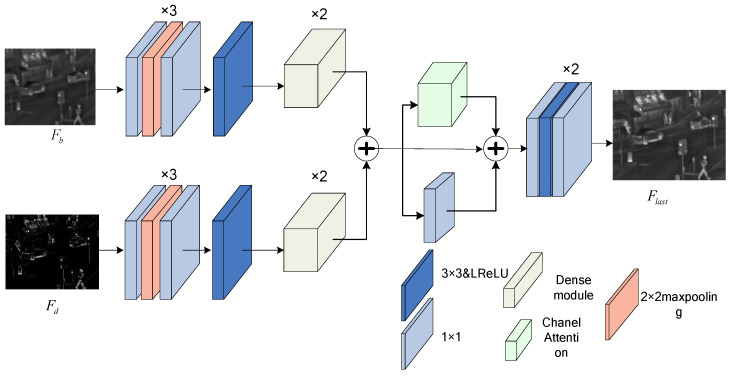
Dense Channel Attention Mechanism Network (dCAMN).

**Figure 6 sensors-24-06916-f006:**
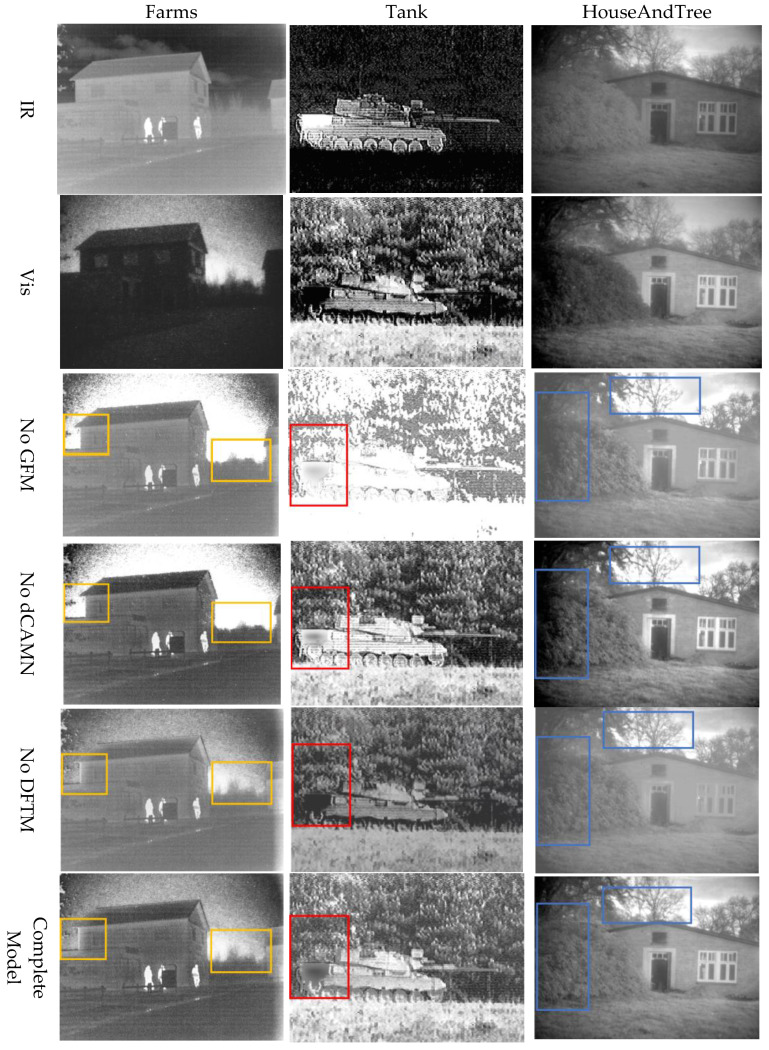
Ablation analysis of our method on the TNO dataset.

**Figure 7 sensors-24-06916-f007:**
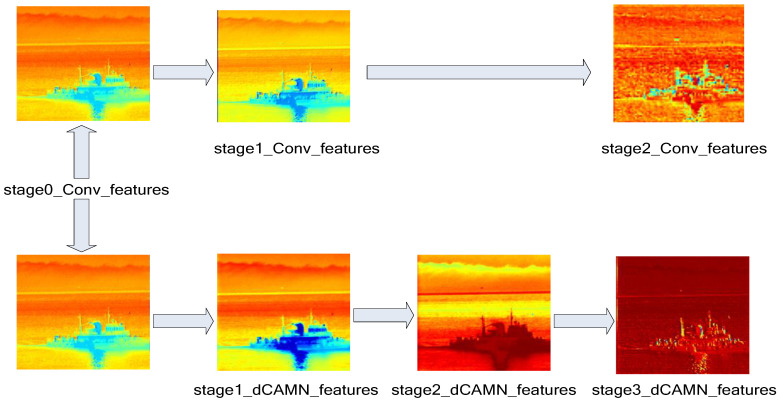
Analysis of the effect of the channel attention mechanism network.

**Figure 8 sensors-24-06916-f008:**
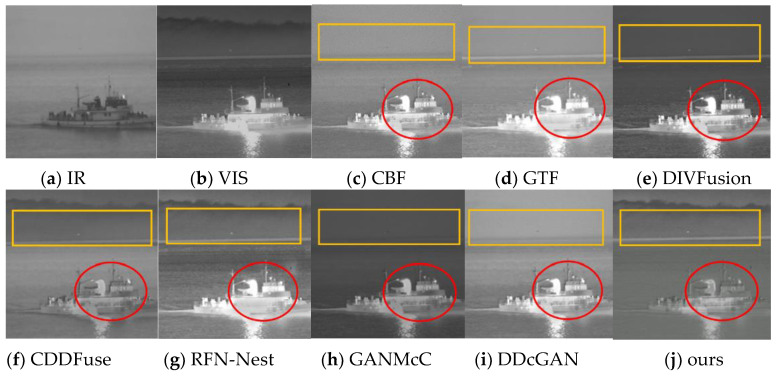
Qualitative fusion results of selected Streamboat scenes in the dataset.

**Figure 9 sensors-24-06916-f009:**
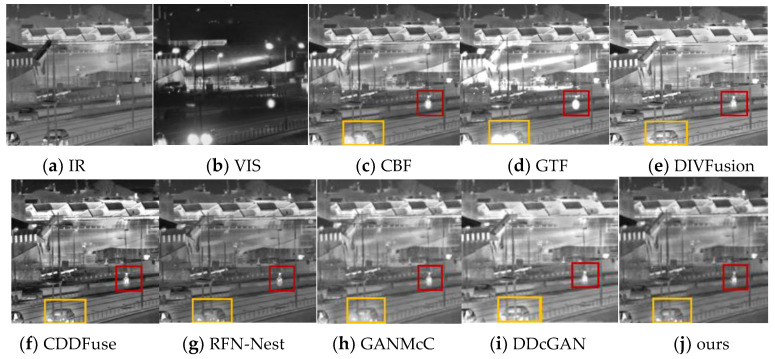
Qualitative fusion results of selected Street scenes in the dataset.

**Figure 10 sensors-24-06916-f010:**
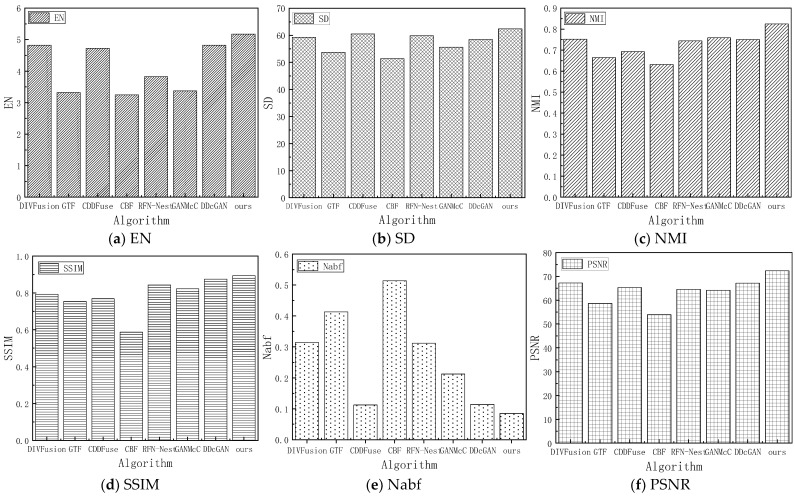
Histogram of indicators of 6 mainstream algorithms in the test set image on Streamboat.

**Figure 11 sensors-24-06916-f011:**
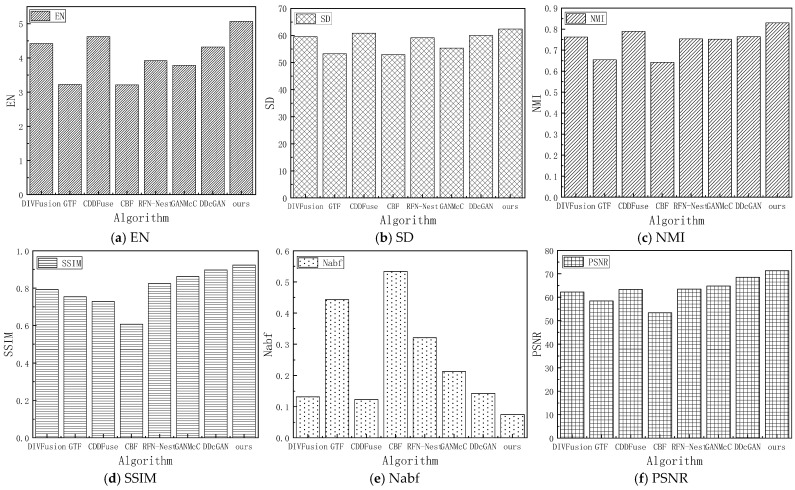
Histogram of indicators of 6 mainstream algorithms in the test set image on Street.

**Figure 12 sensors-24-06916-f012:**
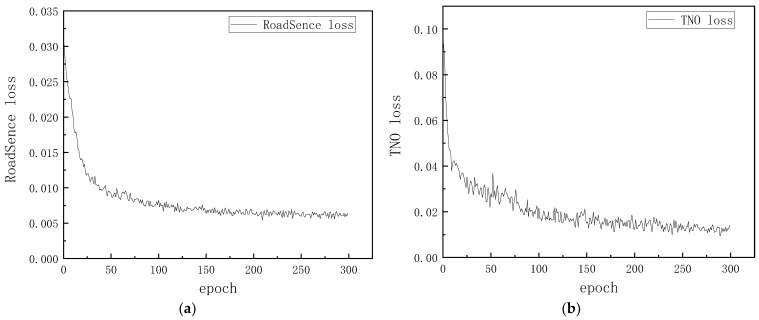
HFCA_GAN trained loss function trend plots are shown in the “RoadSence” and “TNO” datasets. (**a**) Loss function under RoadSence; (**b**) loss function under TNO.

**Table 1 sensors-24-06916-t001:** Ablation experiments on the TNO dataset.

Models	EN	SD	NMI	SSIM	Nabf	PSNR
No GFM	6.463	29.985	0.654	0.795	0.135	62.374
No DFTM	6.534	32.273	0.698	0.865	0.135	61.345
No dCAMN	6.874	30.483	0.764	0.875	0.134	60.094
all	**6.963**	**35.453**	**0.796**	**0.895**	**0.075**	**64.346**

**Table 2 sensors-24-06916-t002:** Ablation experiments on the RoadSence dataset.

Models	EN	SD	NMI	SSIM	Nabf	PSNR
No GFM	6.776	28.434	0.624	0.746	0.143	63.628
No DFTM	6.915	31.467	0.658	0.812	0.197	62.322
No dCAMN	6.977	29.965	0.723	0.864	0.101	58.265
all	**7.285**	**34.264**	**0.821**	**0.891**	**0.081**	**66.591**

**Table 3 sensors-24-06916-t003:** Average EN, SD, NMI, SSMI and Nabf of the TNO image pair.

Metric	CBF	GTF	DIVFusion	CDDFuse	RFN-Nest	GANMcC	DDcGAN	Ours
**EN**	3.137	3.227	4.211	4.875	3.923	3.736	4.355	**5.653**
**SD**	51.238	53.823	57.245	**60.157**	58.322	55.323	58.433	59.932
**NMI**	0.642	0.653	0.683	0.695	0.752	0.735	0.763	**0.794**
**SSIM**	0.582	0.756	0.883	0.890	0.835	0.832	0.874	**0.892**
**Nabf**	0.534	0.443	0.142	0.123	0.356	0.283	0.153	**0.073**
**PSNR**	53.656	58.345	67.854	68.434	63.576	64.865	68.544	**70.543**

**Table 4 sensors-24-06916-t004:** Average EN, SD, NMI, SSMI, Nabf of the RoadSence image pair.

Metric	CBF	GTF	DIVFusion	CDDFuse	RFN-Nest	GANMcC	DDcGAN	Ours
**EN**	3.342	3.587	4.343	4.756	3.873	3.836	4.134	**5.756**
**SD**	50.452	52.473	57.285	**59.137**	57.564	54.653	58.764	58.832
**NMI**	0.723	0.734	0.771	0.736	0.753	0.742	0.752	**0.804**
**SSIM**	0.598	0.726	0.787	0.812	0.826	0.863	0.863	**0.898**
**Nabf**	0.545	0.454	0.121	0.133	0.335	0.212	0.143	**0.067**
**PSNR**	54.754	58.322	67.493	68.532	64.496	67.365	69.432	**71.537**

**Table 5 sensors-24-06916-t005:** Comparison results of computational efficiency of different fusion methods (seconds).

Methods	CBF	GTF	CDDFuse	RFN-Nest	DIVFuison	GANMcC	DDcGAN	Ours
**Roadscene**	1.273	2.987	0.143	0.179	0.104	0.534	0.096	0.043
**TNO**	2.345	3.093	0.253	0.345	0.275	0.794	0.294	0.203

## Data Availability

Publicly available datasets were analyzed in this study.

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
