# Peer review of "Hierarchical Fusion of Infrared and Visible Images Based on Channel Attention Mechanism and Generative Adversarial Networks"

_sensors, 2024, doi:10.3390/s24216916_

Round 1
Reviewer 1 Report
Comments and Suggestions for Authors
1、The comparison algorithms selected are mostly classic, lacking the state-of-the-art or cutting-edge algorithms from recent years for comparison, which casts doubt on the model's leading-edge status.
2、There are errors in the images, such as Fd and Fb in Figure 5.
3、How to obtain the computational efficiency listed in Table 4 needs to be explained in detail.
Comments on the Quality of English LanguageModerate editing of English language required.
Reviewer 2 Report
Comments and Suggestions for Authors
This paper presents a novel hierarchical fusion method for infrared and visible images based on channel attention mechanisms and generative adversarial networks. The proposed approach demonstrates practical value. The authors have conducted comprehensive experiments and provided insightful analysis. While there are some areas that could benefit from minor improvements, the overall quality and contribution of the work are significant. Therefore, I recommend this paper for acceptance after minor revisions.
Minor issues:
1, Figure Improvements:
Figure 7 and Figure 8:
The sub-figures labeled (a) to (j) represent different methods, but the manuscript does not explicitly clarify which method corresponds to each sub-figure. Please clearly label each sub-figure with the corresponding method name in the figure captions to avoid confusion for the readers.Figure 9, Figure 10, and Figure 11: The current quality of these figures is poor, with unclear and blurry text. I recommend redrawing these figures. Additionally, the figure captions should explicitly state which method each sub-figure corresponds to, rather than using generic descriptions like "The indicators (from left to right, top to bottom) are EN, SD, NMI, SSIM, Nabf, and PSNR, respectively." Please specify the methods and metrics for each sub-figure. 2, Reference Update: The references cited in the manuscript are somewhat outdated. I suggest updating the references to include the latest relevant work from 2021-2024.
3, Ablation Studies: It's better to consider adding ablation studies to verify the independent contribution of each module (e.g., guided filtering, histogram mapping, GAN, channel attention mechanism) to the overall performance. This will help us better understand the role of each component.
Reviewer 3 Report
Comments and Suggestions for Authors
11. The paper was well introduced and the problem clearly stated in the introduction section
22. The proposed method was also tested against state of the arts fusion methods in terms of entropy, standard deviation, normalized mutual information, structural similarity, gradient based fusion performance, peak signal-to-noise ratio (PSNR) as well as the computational complexity
3. A related work section is notably missing which should include what other researchers have done relating to fusion methods.
4. The guided filter should also be explained under section 3
5. All Metrics for checking the performance of the proposed method should be explained
6. Which mechanism was used to address overfitting when training the model
7. The abbreviations DTM was not defined before being used
8. The italics used indicate the best performance is not seen in Table 1
9. Figures 8 to 10 should be labelled well
10. Some Text in figures 9 and 10 are blurred and not visible
11. The conclusion should be reframed
12. Check for typographical and grammatical errors
13. Manuscript should also be formatted properly
Comments on the Quality of English Language
Check for typographical and grammatical errors
